# Imaging Tests in the Early Diagnosis of Giant Cell Arteritis

**DOI:** 10.3390/jcm10163704

**Published:** 2021-08-20

**Authors:** Diana Prieto-Peña, Santos Castañeda, Isabel Martínez-Rodríguez, Belén Atienza-Mateo, Ricardo Blanco, Miguel A. González-Gay

**Affiliations:** 1Research Group on Genetic Epidemiology and Atherosclerosis in Systemic Diseases and in Metabolic Bone Diseases of the Musculoskeletal System, IDIVAL, Department of Rheumatology, Hospital Universitario Marqués de Valdecilla, 39008 Santander, Spain; diana.prieto.pena@gmail.com (D.P.-P.); mateoatienzabelen@gmail.com (B.A.-M.); rblancovela@gmail.com (R.B.); 2Department of Rheumatology, Hospital Universitario de La Princesa, IIS-Princesa, 28006 Madrid, Spain; scastas@gmail.com; 3Cátedra UAM-Roche, EPID-Future, Universidad Autónoma Madrid (UAM), 28049 Madrid, Spain; 4Molecular Imaging Group, IDIVAL, Department of Nuclear Medicine, Hospital Universitario Marqués de Valdecilla, University of Cantabria, 39008 Santander, Spain; mimartinez@humv.es; 5School of Medicine, Universidad de Cantabria, 39008 Santander, Spain; 6Cardiovascular Pathophysiology and Genomics Research Unit, School of Physiology, Faculty of Health Sciences, University of the Witwatersrand, Johannesburg 2000, South Africa

**Keywords:** giant cell arteritis, imaging, positron emission tomography, computed tomography, ultrasound

## Abstract

Early recognition of giant cell arteritis (GCA) is crucial to avoid the development of ischemic vascular complications, such as blindness. The classic approach to making the diagnosis of GCA is based on a positive temporal artery biopsy, which is among the criteria proposed by the American College of Rheumatology (ACR) in 1990 to classify a patient as having GCA. However, imaging techniques, particularly ultrasound (US) of the temporal arteries, are increasingly being considered as an alternative for the diagnosis of GCA. Recent recommendations from the European League Against Rheumatism (EULAR) for the use of imaging techniques for large vessel vasculitis (LVV) included US as the first imaging option for the diagnosis of GCA. Furthermore, although the ACR classification criteria are useful in identifying patients with the classic cranial pattern of GCA, they are often inadequate in identifying GCA patients who have the extracranial phenotype of LVV. In this sense, the advent of other imaging techniques, such as magnetic resonance imaging (MRI), computed tomography (CT), and positron emission tomography (PET)/CT, has made it possible to detect the presence of extracranial involvement of the LVV in patients with GCA presenting as refractory rheumatic polymyalgia without cranial ischemic manifestations. Imaging techniques have been the key elements in redefining the diagnostic work-up of GCA. US is currently considered the main imaging modality to improve the early diagnosis of GCA.

## 1. Introduction

Giant cell arteritis (GCA), a condition that often overlaps with polymyalgia rheumatica, is the most common vasculitis among individuals who are over 50 years of age and of northern European ancestry [1,2]. Classically, the diagnosis of GCA is based primarily on the recognition of the cardinal clinical symptoms, including headache, scalp tenderness, jaw claudication, and visual symptoms. The early diagnosis of GCA and the rapid onset of glucocorticoids are essential to avoid the development of serious complications, such as irreversible visual loss [3,4,5]. In 1990, the American College of Rheumatology (ACR) proposed a set of classification criteria for GCA, which included clinical, laboratory, and histopathological findings on temporal artery biopsy [6]. These criteria have been of great help in the identification of patients with the classic cranial symptoms of GCA and, therefore, temporal artery biopsy has been considered, for many years, as the gold standard in confirming the diagnosis of GCA [3]. However, ultrasound (US) of the temporal arteries has emerged as an alternative method to detect inflammatory changes in the vessel wall without the need for biopsy [7,8,9]. In this sense, this imaging technique has progressively gained the support of physicians for use in the early diagnosis of GCA. Furthermore, the advent of other imaging techniques has made it possible to identify the presence of inflammatory vasculitic changes in the aorta and other extracranial vessels of patients with GCA [10,11,12,13,14]. Imaging techniques such as magnetic resonance imaging (MRI), computed tomography (CT), 18F-fluorodeoxyglucose (FDG), positron emission tomography (PET)/CT, and US have been shown to be essential for identifying a subset of patients who present with extracranial involvement without the classic pattern of cranial manifestations of GCA [8,9,15]. Extracranial LVV-GCA often presents with non-specific manifestations, such as refractory polymyalgia rheumatica (PMR) and constitutional symptoms [10,13]. They may present with ischemic extracranial manifestations, mainly with upper or lower limb ischemia [10]. Aneurysms and other severe vascular complications can also be observed. Imaging techniques are crucial for the early diagnosis of this subset of patients.

Based on the need to standardize the use of imaging tools in the diagnosis of LVV, a group of experts from the European League Against Rheumatism (EULAR) has recently provided a set of evidence-based recommendations for the use of imaging in LVV in clinical practice [16]. These recommendations have been of great help for improving the early diagnosis and monitoring of patients with GCA.

Throughout this review, we provide an updated overview of the role of imaging techniques in GCA and their practical application for the early diagnosis of GCA. We included articles published in the PubMed database up to June 2021 using the following search strategies: “Giant Cell Arteritis AND Imaging”, “Giant Cell Arteritis AND Ultrasound”, “Giant Cell Arteritis AND Positron Emission Tomography”, “Giant Cell Arteritis AND Magnetic Resonance Imaging”, “Giant Cell Arteritis AND Computed Tomography Angiography”.

## 2. Ultrasound for GCA Diagnosis

According to the last EULAR recommendations for the use of imaging in LVV [16], US of temporal arteries is recommended as the first imaging tool in patients with suspected predominantly cranial GCA. The four principal US findings in patients with GCA are thickening of the vessel wall or halo sign, non-compressible arteries or compression sign, stenosis, and occlusions.

Since Schmidt et al. [17] reported for the first time the use of US for the diagnosis of GCA in 1997, this technique has been increasingly gaining interest and wide acceptance as a non-invasive and reliable tool for GCA diagnosis [12,18,19,20,21,22,23,24,25,26]. The first meta-analysis conducted by Karassa et al. [18], which included 23 heterogenous primary studies published until 2004, showed modest results for the use of US in GCA diagnosis [25]. Later on, Arida et al. [21] focused on the specific value of the halo sign for GCA diagnosis in a meta-analysis that only included prospective studies that met the technical quality criteria for US. This study described a sensitivity of 68% and a specificity of 91% for the halo sign, which increased to 100% specificity when the bilateral halo sign was present [21]. The most recent studies have reported better results for temporal artery ultrasound, reaching a sensitivity of 91.6% and a specificity of 95.8%, using clinical diagnosis as the reference standard [24]. De Miguel et al. [19] reported excellent inter-reader reliability with kappa values > 0.80. These results were also confirmed by the Outcome Measures in Rheumatology (OMERACT) US LVV group, which found inter-rater agreements of 91–99% and mean kappa values of 0.83–0.98 for both inter-rater and intra-rater reliabilities [26].

Ultrasound in patients with suspected cranial GCA should always include assessment of the temporal and axillary arteries, as stated in the EULAR imaging in LVV recommendations [16]. Temporal artery evaluations should include the common temporal arteries and their frontal and parietal branches, assessed both in longitudinal and transverse planes bilaterally. Other arteries, such as facial, occipital, vertebral, subclavian and femoral arteries, can be examined when the diagnosis of GCA is not clear [16,25,26]. US is not considered the best method for the diagnosis of extracranial GCA. US is a useful method for assessment of the abdominal aorta, but the diagnostic yield in the diagnosis of GCA is limited. Transesophageal echocardiography is a semi-invasive US imaging technique that allows the assessment of the thoracic aorta. It can help in the diagnosis of extracranial GCA in some cases [27].

High-resolution linear probes with color Doppler mode are required in the evaluation of the vasculitis damage. With respect to this, probes with at least ≥15–18 MHz and ≥12–15 MHz frequencies are recommended for the assessment of temporal and axillary arteries, respectively [16]. The Outcome Measures in Rheumatology (OMERACT) group on US for LVV, which included experts from Europe and the USA, was created to reach consensus-based definitions of normal US appearance and key elementary US lesions in suspected GCA. The “halo” and the “compression” signs are the most recognizable US findings for GCA. The halo sign is defined as a homogenous, hypoechoic wall thickening which is well delineated towards the luminal side, visible both in longitudinal and transverse planes, and most commonly concentric in transverse scans (Figure 1). The compression sign should be assessed by applying pressure via the transducer until the lumen of the temporal artery occludes and no arterial pulsation remains visible. The compression sign is positive when the thickened arterial wall remains visible upon compression [26,28].

The increasing improvement of the resolution of US probes has enabled better delineation of the “halo sign” by measuring the intima–media thickness (IMT) in the temporal and axillary arteries. There are still no validated cut-off values for GCA diagnosis, but most sonographers use the cut-off values established by Schäfer et al. in a prospective study conducted using patients with GCA and matched controls [29]. These cut-off values are 0.42, 0.34, 0.29, and 1.0 mm for common temporal arteries, frontal branches, parietal branches, and axillary arteries, respectively [29]. In a further step, van der Geest et al. [30] have recently developed a US composite scoring system, the “Halo Score”, to quantify the extent of vascular inflammation in patients with GCA. The “Halo Score” measures the extent of inflammation in the three segments of temporal arteries and the axillary arteries. The high scores strongly support the diagnosis of GCA and identify patients at high risk of ocular ischemia [30]. There is an ongoing prospective study to assess Halo Score as a diagnostic, prognostic, and monitoring tool for GCA in order to validate these findings [31]. In addition, IMT cut-off values have also been proposed for temporal artery compression sign [32]. In this regard, Czhihal et al. [33] have recently validated a cut-off value of ≥0.7 mm in patients presenting with acute arterial ocular ischemia. However, this study revealed the possible limitations of temporal artery IMT measurements in male patients >70 years of age, in whom specificity and positive predictive value were decreased. The diagnostic accuracy of temporal artery ultrasound appears to be influenced by age, sex, and cardiovascular risk factors. In this regard, de Miguel et al. [34] observed that atherosclerotic disease in the carotid arteries correlated with an increase in temporal artery IMT, which may lead to a false-positive halo. Given that carotid atherosclerosis is present in the majority of patients with suspected GCA, this group proposed a cut-off of temporal artery intima–media thickness > 0.34 mm in at least two branches to minimize false positives in the diagnosis of GCA [34]. Further studies are needed to assess age-, sex-, and cardiovascular risk-adapted cut-off values for temporal artery wall thickness.

US has become the cornerstone of the GCA fast-track clinics as an imaging tool that favors the early diagnosis of GCA [25,35,36]. In these clinics, physicians can refer patients to specialists who can be contacted immediately. Based on clinical and US findings, GCA can be rapidly confirmed or excluded. In case the diagnosis of GCA is unclear, additional tests such as a temporal artery biopsy and/or other imaging techniques should be ordered.

## 3. Magnetic Resonance Imaging for GCA Diagnosis

High-resolution MRI has been shown to be useful for the diagnosis and long-term monitoring of GCA [7,8,16]. Vasculitis on MRI findings presents as increased vessel wall thickness and edema with increased mural enhancement on high-resolution post-contrast images. Currently, it is predominantly used for the assessment of extracranial LVV-GCA. However, recent studies have revealed that MRI could also be useful for cranial GCA [37,38,39,40].

The most recent EULAR guidelines recommended the use of high-resolution 3-T MRI of cranial arteries as an alternative for GCA diagnosis if US is not available or is inconclusive [16]. Several studies have shown the high sensitivity and specificity of MRI for detecting vessel wall inflammation on the cranial arteries of patients with GCA [8,37,38,39,40]. The first studies addressing the role of MRI in cranial GCA diagnosis revealed modest results when compared to clinical diagnosis [39,40]. However, most of these studies used 1T–1.5T MRI machines. Better results were reported in a recent meta-analysis with a pooled sensitivity of 73% and a specificity of 88% for MRI in comparison to clinical GCA diagnosis and a sensitivity of 93% and a specificity of 81% when temporal artery biopsy was considered as the reference standard [8]. A recent cross-sectional study has compared the performance of MRI to US for detecting vasculitis in 35 patients with new-onset or already diagnosed GCA [41]. No statistical differences for detecting vasculitic changes in temporal arteries were observed between the two imaging techniques, except for the frontal artery where MRI was superior to US. It is of note that a modest interobserver agreement was observed (kappa 0.44) for MRI interpretation in comparison to US, which showed excellent results. Sommer et al. [42] assessed the potential role of three-dimensional high-resolution contrast enhanced black blood MRI in determining the arteritic nature of anterior ischemic optic neuropathy (AION) in 27 patients with suspected GCA. This technique detected arteritis involvement of the posterior ciliary arteries earlier than fundoscopy in some patients. The authors proposed that MRI may be useful in patients with suspected GCA with visual impairment but unremarkable fundoscopy, in order to rule out arteritic AION [42].

The main advantage of MRI for cranial GCA diagnosis is the possibility of assessing multiple cranial arteries at the same time. MRI may play a special role in the evaluation of cranial nerves in patients with GCA, where it has shown excellent resolution [43]. However, its use is limited in some centers due to the high cost and low availability of 3T MRI machines. In addition, some conditions, such as renal failure, implanted devices, or claustrophobia contraindicate the use of MRI. Taking all these considerations into account, US remains the imaging modality of choice for the assessment of cranial GCA.

Regarding the role of MRI in the assessment of extracranial vessel inflammation, it has been included in the EULAR recommendations both for supporting the diagnosis of LVV-GCA and for long-term monitoring of structural damage, particularly to detect stenosis, occlusions, and aneurysms [16]. The choice of MRI over US, CT, or PET/CT scan for patients with LVV-GCA depends on local availability and expertise at each individual center. Most evidence of the usefulness of MRI in LVV comes from studies on Takayasu’s arteritis. However, no specific studies on LVV-GCA have been performed [8,9].

## 4. Computed Tomography Angiography for GCA Diagnosis

CTA is another option for the diagnosis of extracranial LVV-GCA. Mural thickening with double ring enhancement after an intravenous injection of iodine-based contrast is observed in patients with LVV. Several studies have compared CTA with PET/CT for the diagnosis of LVV [44,45,46,47,48]. De Boysson et al. reported 95% sensitivity and 100% specificity for CTA using PET/CT as a reference [45]. Moragas-Solanes et al. [48] conducted a retrospective study on 59 patients with clinical suspicion of LVV who underwent PET/CT and CTA. A higher sensitivity for PET/CT (95.6%) compared to CTA (60.9%) was observed for LVV diagnosis. Vaidyanathan et al. found similar results in a series of 36 patients comparing the accuracy of PET/CT and CTA [47]. The area under the curve (AUC) for SUVmax on PET/CT was 0.95, and for mural thickening on CTA it was 0.83 [47]. These results suggest a greater potential of PET/CT for the detection and extension of LVV-GCA.

## 5. 18F-FDG (Fluorodeoxyglucose)-Positron Emission Tomography (PET)/Computed Tomography (CT) for GCA Diagnosis

18F-FDG-PET combined with CT is a functional imaging technique that has demonstrated usefulness for LVV diagnosis due to its ability to detect glucose uptake from the high activity of inflammatory cells in the vessel walls. PET/CT yields an exceptional overview of the extension of vascular inflammation. In addition, it is useful to rule out other entities such as malignancy or infection (Figure 2) [7,8,14,49].

The acquisition and interpretation of FDG-PET/CT images for LVV diagnosis can be challenging. For this reason, the Society of Nuclear Medicine and Molecular Imaging (SNMMI), the European Association of Nuclear Medicine (EANM), and the PET Interests Group endorsed by the American Society of Nuclear Cardiology (ASNC) provided in 2018 joint procedural recommendations on FDG-PET/CT imaging for LVV [49]. Glucocorticoids may reduce vascular wall uptake of FDG and increase FDG uptake in the liver, leading to underestimation of vascular FDG uptake. For this reason, it is recommended to stop or delay glucocorticoid therapy whenever possible. Consensus-based recommendations for the correct acquisition of FDG-PET/CT images include a fasting period for at least 6 h before FDG administration, glucose levels below 126–160 mg/dL, withdrawal, or delay of glucocorticoid therapy whenever possible, and a minimum interval of 60 min between FDG administration and acquisition to assure adequate biodistribution. Some experts recommend a preferable interval of 180 min after FDG injection in order to provide a better delineation of the aortic wall uptake [50,51], while the most recent study suggested an interval of 120 min [52].

Regarding the interpretation and reporting of FDG-PET/CT, experts recommend the use of a standardized 0-to-3 visual grading scale as follows: 0 = no uptake (≤mediastinum); 1 = low-grade uptake (<liver); 2 = intermediate-grade uptake (=liver); 3 = high-grade uptake (>liver), with grade 2 possibly indicative of and grade 3 considered positive for active LVV [49]. A total vascular score can be calculated by summing up the grade of uptake at different vascular areas.

In addition, semiquantitative methods have also been proposed for research purposes or to assess therapy response [49]. The target-to-background ratio (TBR) is the recommended method. TBR can be calculated by dividing the arterial wall maximum standardized uptake value (SUV_max_) by the blood pool SUV_max_ or liver SUV_max_. Normalization of the arterial wall uptake to the background activity of the blood pool provides a good reference for assessing vascular inflammation [42]. In this regard, our group conducted a prospective study on 43 LVV patients to establish a threshold index for application in the clinical setting. Semiquantitative analysis of PET/CT images acquired 180 min after F-FDG injection and the TBR index of 1.34 showed very high accuracy (sensitivity—100%, specificity—94.4%) for the detection of LVV [51].

The 2018 EULAR imaging guidelines for LVV [14] include PET/CT as an imaging modality of choice for LVV-GCA diagnosis, based on the results of several meta-analyses that confirmed its diagnostic accuracy [53,54,55]. The most recent meta-analysis was conducted by Lee et al. [55] and included eight studies involving 400 subjects. The pooled sensitivity and specificity for LVV-GCA diagnosis were 83.3% and 89.6%, respectively, with an AUC of 0.884.

In current clinical practice, PET/CT is being increasingly used for the early diagnosis of isolated extracranial LVV-GCA. This technique has been shown to be particularly useful in patients who present with refractory polymyalgia rheumatica, associated with atypical symptoms such as a predominant pelvic girdle involvement, inflammatory lumbar pain, and limb claudication [15].

Currently, the use of PET/CT is not indicated for patients with predominant cranial symptoms of GCA due to the difficulty of detecting inflammatory changes at the cranial arteries even for experienced clinicians [14,56,57]. However, recent studies have also shown the potential role of PET/CT in the diagnosis of cranial GCA [58,59]. Sammel et al. [59] conducted a prospective study on 64 patients with newly suspected GCA who underwent both PET/CT and temporal artery biopsy within the first 72 h of starting glucocorticoid therapy. The sensitivity of PET/CT was 92% and the specificity was 85%, using temporal artery biopsy as a reference. When compared to clinical diagnosis, PET/CT had a sensitivity of 71% and a specificity of 91%.

Some of the limitations of this imaging technique are the high costs and the exposure to high levels of radiation. Another potential limitation is the decreased diagnostic accuracy of PET/CT in patients receiving high-dose glucocorticoid therapy. In this regard, some authors recommend performing PET/CT scans within the first 10 days of treatment with high-dose glucocorticoids [60]. Atherosclerosis has also been considered a pitfall in the interpretation of PET/CT. Although a distinctive diffuse pattern of vascular FDG uptake is observed in patients with LVV, while atherosclerosis usually shows a focal or “patchy” uptake, distinguishing atherosclerosis and LVV can be challenging [61,62]. Experts recommend the use of semiquantitative measurements and cumulative vascular scores for PET/CT interpretation, and the combined use of PET and CTA whenever possible in order to avoid false-positive results [62].

## 6. Discussion

Early diagnosis of GCA is crucial to improve outcomes for GCA patients and prevent irreversible vascular damage. In this sense, imaging tests have greatly advanced in the treatment of GCA in recent years. Among them, US of the temporal arteries is now considered the first tool to evaluate patients with suspected GCA. This rapid access imaging technique allows non-invasive identification of patients with cranial manifestations who are at increased risk of developing blindness and other serious vascular complications [8,9,16,25,35,36,61]. US can also be useful for the evaluation of extracranial vessels such as the axillary arteries. However, its limitations in evaluating vascular inflammatory changes in the thoracic and abdominal aorta make other imaging techniques more suitable for this purpose. In this sense, MRI, CTA, or PET/CT are excellent tools to identify patients with extracranial LVV-GCA, in whom the diagnosis is often delayed due to the non-specific manifestations [8,9,16]. In this sense, it is not uncommon to identify LVV involvement in patients in whom a diagnosis of polymyalgia rheumatica is initially established due to the clinical absence of ischemic manifestations of GCA [63,64].

In practical terms, we strongly support the use of temporal artery US for the evaluation of patients with predominant cranial manifestations of GCA. It allows a diagnosis of GCA to be confirmed without much delay. However, when the clinical suspicion of GCA is high and the US is not positive, other diagnostic tests, such as temporal artery biopsy, should be considered.

The diagnosis of patients with predominant extracranial GCA should be supported by the use of imaging techniques such as US, CT, MRI, and/or PET/CT. In those patients in whom the temporal artery biopsy yield is low, the imaging modality of choice will depend on the availability and experience in each individual center. In our hands, US of the axillary and other extracranial arteries is a good imaging tool for the preliminary evaluation of extracranial LVV-GCA. Nevertheless, due to the limitations of US in visualizing deep vessels, we prefer to perform MRI, CT, or PET/CT scans in patients in whom the clinical suspicion of extracranial involvement of LVV-GCA is high, particularly when there are ischemic manifestations such as limb claudication. These techniques provide an excellent overview of the extension of extracranial LVV-GCA. We have obtained good results in identifying LVV by PET/CT scan in patients with constitutional symptoms without ischemic manifestations [15]. Using this technique also allows us to exclude other conditions such as malignancy (Figure 3).

It should be noted that imaging test results should always be interpreted in the context of the level of suspicion of GCA, which is generally based on the clinical judgment and experience of the physician treating patients with GCA. Efforts have been made by different expert groups to develop pre-test clinical probability scores for GCA to help clinicians estimate the likelihood of GCA [65,66]. These scores can be potentially useful in ensuring a more homogeneous management of patients with GCA and in optimizing the performance of imaging techniques.

In conclusion, the use of imaging techniques has revolutionized the diagnosis of GCA, making it possible to improve the early diagnosis of GCA. In particular, temporal artery US has become the main imaging technique in the clinical evaluation and treatment of GCA, and it is currently used in most centers treating patients with GCA.

## Figures and Tables

**Figure 1 jcm-10-03704-f001:**
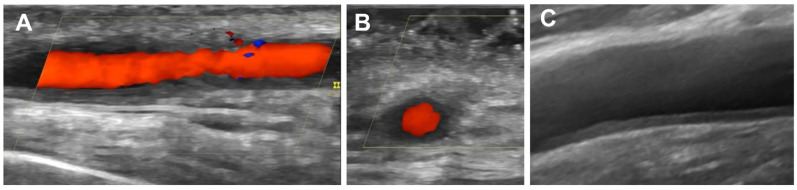
“Halo sign” on temporal and axillary arteries: (**A**) longitudinal view of the temporal parietal artery branch; (**B**) transverse view of the temporal parietal artery branch; (**C**) longitudinal view of the axillary artery.

**Figure 2 jcm-10-03704-f002:**
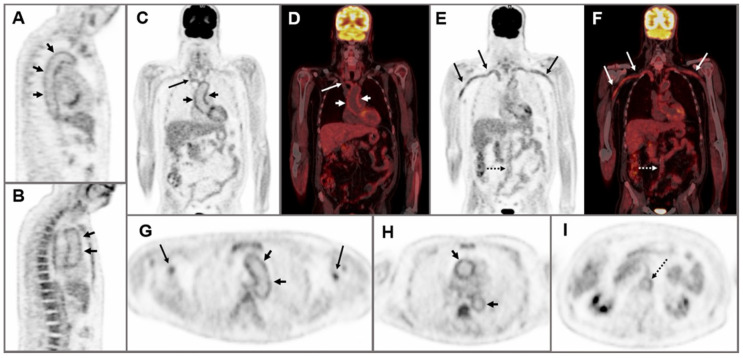
A 63-year-old man with a 4-month history of right hemicranial headache, scalp hyperalgesia, asthenia, weight loss, and increased erythrocyte sedimentation rate and serum C-reactive protein levels. Sagittal PET (**A**,**B**), coronal PET and fused PET/CT (**C**–**F**), and axial PET images (**G**–**I**) showed hypermetabolism along the vessel wall of the brachiocephalic trunk, subclavian and axillary arteries (arrows), thoracic aorta (short arrows) and abdominal aorta (dotted arrow), suggesting large vessel vasculitis. Temporal artery biopsy was positive for giant cell arteritis.

**Figure 3 jcm-10-03704-f003:**
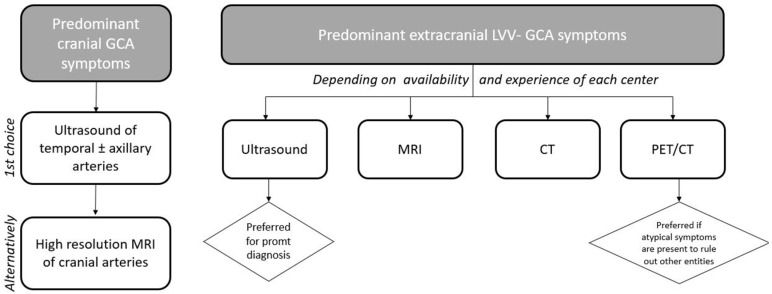
Algorithm for the use of imaging techniques in giant cell arteritis in clinical practice.

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
