# Peer review of "Imaging Tests in the Early Diagnosis of Giant Cell Arteritis"

_jcm, 2021, doi:10.3390/jcm10163704_

Round 1

Reviewer 1 Report

Interesting paper however abstract and publication unprecise in regard to what is the issue of importance in this review - US?, all modalities?

The review in reference no 42 covers most of the current new review so what is  new and interesting? The new information should focus on US as an important technique in the clinical evaluation and diagnostic work-up of GCA and LVV vasculitis.

Advice for a revision to achieve a focus for the relevance of this new review, with emphasis and focus on US as an important new imaging technique in the work-up of GCA and LVV conditions..

Author Response

Manuscript ID: JCM-1295597

IMAGING TESTS IN THE EARLY DIAGNOSIS OF GIANT CELL ARTERITIS

We are pleased to upload the Revised manuscript. Please, find below a detailed point-by-point response to the Reviewers. Changes have been marked up using the “Track Changes” function in the manuscript.

Reviewer 2

Interesting paper however abstract and publication unprecise in regard to what is the issue of importance in this review - US?, all modalities?

The review in reference no 42 covers most of the current new review so what is new and interesting? The new information should focus on US as an important technique in the clinical evaluation and diagnostic work-up of GCA and LVV vasculitis.

Advice for a revision to achieve a focus for the relevance of this new review, with emphasis and focus on US as an important new imaging technique in the work-up of GCA and LVV conditions..

We thank the Reviewer for prompting us to highlight the relevance of this review. Our aim was to include relevant information of all imaging modalities that are currently used for the diagnosis of GCA with a focus on their practical application for an early diagnosis of GCA. We agree that ultrasound is now considered the most important imaging technique for GCA diagnosis and therefore we have highlighted this aspect in the manuscript as follows:

Abstract (Page1, line 30):

“Imaging techniques have been the key elements to redefine the diagnostic work-up of GCA. US is currently considered the main imaging modality to improve the early diagnosis of GCA”

Introduction (Page 2, line 68):

Throughout this review, we provide an updated overview of the role of imaging techniques in GCA and their practical application for the early diagnosis of GCA”

Discussion (Page 8, line 360):

“In conclusion, the use of imaging techniques has revolutionized the diagnosis of GCA, making it possible to improve the early diagnosis of GCA. In particular, temporal artery US has become the main imaging technique in the clinical evaluation and treatment of GCA, which is currently used in most centers treating patients with GCA.”

Reviewer 2 Report

This review article is well written and adequately referenced. Some issues warrant consideration:

Abstract/Introduction/Discussion:

- Aneuryms are late sequelae of GCA and thus are usually not found by imaging at early diagnosis of GCA. Moreover, it is not proven that early diagnosis prevents development of aneurysms in the long term. By contrast, early diagnosis prevents early ischemic complications. This should be clarified.

- As ultrasound imaging is multimodal in the diagnosis of GCA, the term Doppler ultrasound should be avoided.

- All imaging techniques mentioned are really not new…

- Tests of imaging results should always be interpreted in the context of clinical     probability. The article should reflect recent efforts on explicite pretest probability assessment in GCA.

Ultrasound:

- Carotid atherosclerosis is present in the majority of patients with suspected GCA.

- US is a useful method for assessment of the abdominal aorta, but the  diagnostic yield in the diagnosis of GCA is limited. Transesophageal    echocardiography may aid in the diagnosis of extracranial GCA in some cases. The sentence on sonographic assessment of the aorta should be more precise.

- Aschwanden et al. were the first to report compression sonography of the   temporal arteries. Their original article should be cited (PMID 22693039).

- A recent paper revealing the limitations of temporal artery IMT measurements in patients presenting with ocular ischemia may be of value (PMID 33123722)

- The B-mode image depicted in Fig. 1C is insufficient; the IMT is not visible as the vessel is not displayed parallel to the skin surface.

- Ultrasound in suspected cranial GCA always should include assessment of the temporal and axillary arteries, as stated in the EULAR imaging recommendations.

MRI:

- A publications proposing a potential role of contrast enhanced black blood MRI at 3T in determining the arteritic nature of anterior ischemic optic neuropathy may be of value (PMID 30095558).

CTA/PET-CT

 - The EANM recommendation to stop or delay glucocorticoid treatment whenever possible, should be commented critically.

- A more recent study suggested a 120 min interval between FDG-injection and PET-scan (PMID 31645635).

- Monitoring of disease activity seems to be beyond the scope of an article focussing on early diagnosis of GCA.

Author Response

We are pleased to upload the Revised manuscript. Please, find below a detailed point-by-point response to the Reviewers. Changes have been marked up using the “Track Changes” function in the manuscript.

Reviewer 1

  1. Aneuryms are late sequelae of GCA and thus are usually not found by imaging at early diagnosis of GCA. Moreover, it is not proven that early diagnosis prevents development of aneurysms in the long term. By contrast, early diagnosis prevents early ischemic complications. This should be clarified.

We agree that early diagnosis of GCA has been shown to prevent the development of ischemic complications, whereas aneurysms are generally found in later stages of the disease. To avoid confusion, we have modified these phrases as follows:

Abstract (Page 1, line 17)

“Early recognition of giant cell arteritis (GCA) is crucial to avoid the development of ischemic vascular complications, such as blindness. aneurysms, and stenosis

  1. As ultrasound imaging is multimodal in the diagnosis of GCA, the term Doppler ultrasound should be avoided.

Following the Reviewer suggestion, the term Doppler has been removed.

Abstract (Page 1, line 20)

“However, imaging techniques, particularly Doppler ultrasound (US) of the temporal arteries are increasingly being considered as an alternative for the diagnosis of GCA.”

  1. All imaging techniques mentioned are really not new…

We agree that the imaging techniques mentioned throughout the manuscript have been used for a long time, although most of them were used initially for other clinical purposes and more recently for the diagnosis of large vessel vasculitis. To be more precise, the term "new" has been removed.

Abstract (Page 1, line 26)

“In this sense, the advent of other new imaging techniques, such as magnetic resonance imaging (MRI), computed tomography (CT) and positron emission tomography.”

Introduction (Page 2, line 50)

“Furthermore, the advent of other new imaging techniques made it possible to identify the presence of inflammatory vasculitic changes in the aorta and other extracranial vessels of patients with GCA.”

  1. Tests of imaging results should always be interpreted in the context of clinical The article should reflect recent efforts on explicite pretest probability assessment in GCA.

We thank the Reviewer for raising this interesting point. We have now included a brief paragraph on pretest probability scores for GCA. Two references have also been added to address this topic.

Discussion (page 7, line 333)

“It should be noted that imaging test results should always be interpreted in the context of the level of suspicion for GCA, which is generally based on the clinical judgment and experience of the physician treating patients with GCA. Efforts have been made by different expert groups to develop pre-test clinical probability scores for GCA to help clinicians estimate the likelihood of GCA [62,63]. These scores can be potentially useful to ensure a more homogeneous management of patients with GCA and to optimize the performance of imaging techniques.”

[62] Czihal M, Lottspeich C, Bernau C, Henke T, Prearo I, Mackert M et al. A Diagnostic Algorithm Based on a Simple Clinical Prediction Rule for the Diagnosis of Cranial Giant Cell Arteritis. J Clin Med. 2021;10:1163. doi: 10.3390/jcm10061163. PMID: 33802092; PMCID: PMC8001831.

[63] Sebastian A, Tomelleri A, Kayani A, Prieto-Pena D, Ranasinghe C, Dasgupta B. Probability-based algorithm using ultrasound and additional tests for suspected GCA in a fast-track clinic. RMD Open. 2020;6:e001297. doi: 10.1136/rmdopen-2020-001297. PMID: 32994361; PMCID: PMC7547539.

  1. Carotid atherosclerosis is present in the majority of patients with suspected GCA.

We thank the Reviewer for prompting us to clarify this issue. The presence of atherosclerosis in the carotid arteries has been related to an increased intima-media thickness in the temporal arteries which can lead to misinterpretation of ultrasound findings. We aimed to highlight this potential limitation of temporal artery US interpretation. We have modified the paragraph addressing this topic to better clarify this interesting point:

Ultrasound for GCA diagnosis (Page 4, line 149):

“In this line, de Miguel et al. [32] observed that atherosclerotic disease in the carotid arteries correlated with an increase in temporal artery IMT, which may lead to a false-positive halo. Given that carotid atherosclerosis is present in the majority of patients with suspected GCA, this group proposed a cut-off of temporal artery intima-media thickness > 0.34 mm in at least two branches to minimize false positives in the diagnosis of GCA [32]. Further studies are needed to assess age-, sex- and cardiovascular risk-adapted cut-off values for temporal artery wall thickness.”

  1. US is a useful method for assessment of the abdominal aorta, but the diagnostic yield in the diagnosis of GCA is limited. Transesophageal echocardiography may aid in the diagnosis of extracranial GCA in some cases. The sentence on sonographic assessment of the aorta should be more precise.

We thank the Reviewer for allowing us to better explain this issue. For this purpose, we have discussed the role of US for the assessment of abdominal aorta in more detail.

Ultrasound for GCA diagnosis (Page 3, line 104):

“US is not considered the best method for the diagnosis of extracranial GCA. US is a useful method for assessment of the abdominal aorta, but the diagnostic yield in the diagnosis of GCA is limited. Transesophageal echocardiography is a semi-invasive US imaging technique that allows the assessment of the thoracic aorta. It can help in the diagnosis of extracranial GCA in some cases [25].”

Discussion (Page 7, line 306)

“US can also be useful for the evaluation of extracranial vessels such as the axillary arteries. However, its limitations in evaluating vascular inflammatory changes in the thoracic and abdominal aorta make other imaging techniques more suitable for this purpose.”

  1. Aschwanden et al. were the first to report compression sonography of the temporal arteries. Their original article should be cited (PMID 22693039).

We agree that the original article must be cited, which has been now included as reference 26.

[26] Aschwanden M, Daikeler T, Kesten F, Baldi T, Benz D, Tyndall A et al. Temporal artery compression sign--a novel ultrasound finding for the diagnosis of giant cell arteritis. Ultraschall Med. 2013;34:47-50. doi: 10.1055/s-0032-1312821. PMID: 22693039.

  1. A recent paper revealing the limitations of temporal artery IMT measurements in patients presenting with ocular ischemia may be of value (PMID 33123722).

We thank the Reviewer for prompting us to discuss in further detail the limitations of temporal artery IMT measurements based on this interesting study.

Ultrasound for GCA diagnosis (Page 3, line 142 and following):

“In addition, IMT cut-off values ​​have also been proposed for temporal artery compression sign [30]. In this regard, Czihal et al. [31] have recently validated a cut-off value of ≥ 0.7 mm in patients presenting with acute arterial ocular ischemia. However, this study revealed possible limitations of temporal artery IMT measurements in male patients> 70 years of age in whom specificity and positive predictive value were decreased. The diagnostic accuracy of temporal artery ultrasound appears to be influenced by age, sex, and cardiovascular risk factors.”

[30] Czihal M, Schröttle A, Baustel K, Lottspeich C, Dechant C, Treitl KM et al. B-mode sonography wall thickness assessment of the temporal and axillary arteries for the diagnosis of giant cell arteritis: a cohort study. Clin Exp Rheumatol. 2017;35:128-133. PMID: 28375835.

[31] Czihal M, Köhler A, Lottspeich C, Prearo I, Hoffmann U, Schulze-Koops H et al. Temporal artery compression sonography for the diagnosis of giant cell arteritis in elderly patients with acute ocular arterial occlusions. Rheumatology. 2021;60:2190-2196. doi: 10.1093/rheumatology/keaa515. PMID: 33123722.

  1. The B-mode image depicted in Fig. 1C is insufficient; the IMT is not visible as the vessel is not displayed parallel to the skin surface.

We have modified Fig. 1C as recommended by the Reviewer.

  1. Ultrasound in suspected cranial GCA always should include assessment of the temporal and axillary arteries, as stated in the EULAR imaging recommendations.

We have highlighted this aspect in the manuscript as follows:

Ultrasound for GCA diagnosis (Page 2, line 97):

“Ultrasound in patients with suspected cranial GCA always should include assessment of the temporal and axillary arteries, as stated in the EULAR imaging in LVV recommendations [14].”

  1. A publication proposing a potential role of contrast enhanced black blood MRI at 3T in determining the arteritic nature of anterior ischemic optic neuropathy may be of value (PMID 30095558).

We agree this study deserves to be mentioned and has now been referred in the manuscript as follows:

Magnetic Resonance Imaging for GCA diagnosis (Page 4, line 183):

“Sommer et al. [40] assessed the potential role of three-dimensional high-resolution contrast enhanced black blood MRI in determining the arteritic nature of anterior ischemic optic neuropathy in 27 patients with suspected GCA. This technique detected arteritis involvement of the posterior ciliary arteries earlier than fundoscopy in some patients. The authors proposed that MRI may be useful in patients with suspected GCA with visual impairment but unremarkable fundoscopy to rule out arteritic AION [40].”

[40] Sommer NN, Treitl KM, Coppenrath E, Kooijman H, Dechant C, Czihal M et al. Three-Dimensional High-Resolution Black-Blood Magnetic Resonance Imaging for Detection of Arteritic Anterior Ischemic Optic Neuropathy in Patients With Giant Cell Arteritis. Invest Radiol. 2018;53:698-704. doi: 10.1097/RLI.0000000000000500. PMID: 30095558.

  1. The EANM recommendation to stop or delay glucocorticoid treatment whenever possible, should be commented critically.

We have now highlighted this recommendation following the Reviewer recommendation.

F-FDG (Fluorodeoxyglucose)-Positron Emission Tomography (PET)/ Computed To-mography (CT) for GCA diagnosis (Page 6, line 236):

“Glucocorticoids may reduce vascular wall uptake of FDG and increase FDG uptake in the liver leading to underestimation of vascular FDG uptake. For this reason, it is recommended to stop or delay glucocorticoid therapy whenever possible.”

  1. A more recent study suggested a 120 min interval between FDG-injection and PET-scan (PMID 31645635).

This interesting annotation has now been included as follows:

F-FDG (Fluorodeoxyglucose)-Positron Emission Tomography (PET)/ Computed To-mography (CT) for GCA diagnosis (Page 6, line 244):

“Some experts recommend a preferable interval of 180 minutes after FDG injection to provide a better delineation of the aortic wall uptake [48,49], while a most re-cent study suggested an interval of 120 minutes [50]”

[50] Rosenblum JS, Quinn KA, Rimland CA, Mehta NN, Ahlman MA, Grayson PC. Clinical Factors Associated with Time-Specific Distribution of 18F Fluorodeoxyglucose in Large-Vessel Vasculitis. Sci Rep. 2019;9:15180. doi: 10.1038/s41598-019-51800-x. PMID: 31645635; PMCID: PMC6811531.

  1. Monitoring of disease activity seems to be beyond the scope of an article focussing on early diagnosis of GCA.

We agree that the role of imaging techniques for monitoring disease activity in GCA might be beyond the scope of this article. Therefore, the paragraphs addressing this issue have been removed.

Round 2

Reviewer 1 Report

Substantial and relevant update of previous article, however there are still a large amount of self-citations and would advocate for a adding relevant references for example at line 167.

Line 167: Lack of references - is it because of relations to the next section?

Line 38: GCA is not the most common vasculitis - Polymyalgia Rheumatica is app. double in prevalance? Suggestions of correction of this line and adding more and more recent references ? (fx: Sharma A et al; doi: 10.1016 /j.semarthrit.2020.07.005).

Line 39-43: all references self-citations. Should be updated. For example: Emamifar A et al 2020: doi:10.1002/acr2.11163

Line 272/section:  Revision OK but should keep in mind that PET-CT is very difficult in use to assess cranial involvement of GCA even for experienced clinicians. False positive findings etc. Suggest adding references for example: Dunpny MP et al, J Nucl Med 2005 or Ben-Haim S et al, J Nucl Med, 2005 and  Emamifar A et al; 2020 doi:10.1002/acr2.11163.

Line 275: 3 references are not correct regarding text and numbers after extensive text revision.

Author Response

Manuscript ID: JCM-1295597

IMAGING TESTS IN THE EARLY DIAGNOSIS OF GIANT CELL ARTERITIS

Reviewer 1

Substantial and relevant update of previous article, however there are still a large amount of self-citations and would advocate for a adding relevant references for example at line 167.

Line 167: Lack of references - is it because of relations to the next section?

The reviewer is right. References belonging to line 167 are related to the next section. We have now included references 37 to 40 in line 167 as well.

Magnetic Resonance Imaging for GCA diagnosis (line 164-168)

“High-resolution MRI has shown to be useful for the diagnosis and long-term monitoring of GCA [7–9,16]. Vasculitis on MRI findings presents as increased vessel wall thickness and edema with increased mural enhancement on high-resolution post-contrast images. Currently, it is predominantly used for the assessment of extra-cranial LVV-GCA. However, recent studies revealed that MRI could also be useful for cranial GCA [37-40].”

[37]     Bley TA, Uhl M, Carew J, Markl M, Schmidt D, Peter H-H, et al. Diagnostic value of high-resolution MR imaging in giant cell arteritis. AJNR Am J Neuroradiol 2007;28:1722–7. https://doi.org/10.3174/ajnr.A0638.

[38]     Geiger J, Bley T, Uhl M, Frydrychowicz A, Langer M, Markl M. Diagnostic value of T2-weighted imaging for the detection of superficial cranial artery inflammation in giant cell arteritis. J Magn Reson Imaging JMRI 2010;31:470–4. https://doi.org/10.1002/jmri.22047.

[39]     Klink T, Geiger J, Both M, Ness T, Heinzelmann S, Reinhard M, et al. Giant cell arteritis: diagnostic accuracy of MR imaging of superficial cranial arteries in initial diagnosis-results from a multicenter trial. Radiology 2014;273:844–52. https://doi.org/10.1148/radiol.14140056.

[40]     Rhéaume M, Rebello R, Pagnoux C, Carette S, Clements-Baker M, Cohen-Hallaleh V, et al. High-Resolution Magnetic Resonance Imaging of Scalp Arteries for the Diagnosis of Giant Cell Arteritis: Results of a Prospective Cohort Study. Arthritis Rheumatol Hoboken NJ 2017;69:161–8. https://doi.org/10.1002/art.39824.

Line 38: GCA is not the most common vasculitis - Polymyalgia Rheumatica is app. double in prevalance? Suggestions of correction of this line and adding more and more recent references? (fx: Sharma A et al; doi: 10.1016 /j.semarthrit.2020.07.005).

We have modified this line to include that giant cell arteritis often overlaps with polymyalgia rheumatica which is the most common vasculitis in elderly patients, as mentioned in the study suggested by the reviewer (Sharma A et al.) that has now been included in the introduction.

Introduction (line 38-40)

“Giant cell arteritis (GCA), a condition that often overlaps with polymyalgia rheumatica, is the most common vasculitis among individuals over 50 years of northern European ancestry [1,2].”

[2] Sharma A, Mohammad AJ, Turesson C. Incidence and prevalence of giant cell arteritis and polymyalgia rheumatica: A systematic literature review. Semin Arthritis Rheum. 2020;50:1040-1048. doi: 10.1016/j.semarthrit.2020.07.005.

Line 39-43: all references self-citations. Should be updated. For example: Emamifar A et al 2020: doi:10.1002/acr2.11163

References have been updated as suggested by the reviewer. The study by Emamifar A et al has now been included as reference number [14]:

[14] Emamifar A, Ellingsen T, Hess S, Gerke O, Hviid Larsen R, Ahangarani Farahani Z, et al. The Utility of 18F-FDG PET/CT in Patients With Clinical Suspicion of Polymyalgia Rheumatica and Giant Cell Arteritis: A Prospective, Observational, and Cross-sectional Study. ACR Open Rheumatol. 2020;2:478-490. doi: 10.1002/acr2.11163.

Line 272/section:  Revision OK but should keep in mind that PET-CT is very difficult in use to assess cranial involvement of GCA even for experienced clinicians. False positive findings etc. Suggest adding references for example: Dunpny MP et al, J Nucl Med 2005 or Ben-Haim S et al, J Nucl Med, 2005 and  Emamifar A et al; 2020 doi:10.1002/acr2.11163.

Following the reviewer suggestion we have highlighted this interesting aspect in the manuscript.

18F-FDG PET/CT for GCA diagnosis (line 274)

“Currently, the use of PET/CT is not indicated for patients with predominant cranial symptoms of GCA due to the difficulty to detect inflammatory changes at the cranial arteries even for experienced clinicians [14,56,57].”

[14] Emamifar A, Ellingsen T, Hess S, Gerke O, Hviid Larsen R, Ahangarani Farahani Z, et al. The Utility of 18F-FDG PET/CT in Patients With Clinical Suspicion of Polymyalgia Rheumatica and Giant Cell Arteritis: A Prospective, Observational, and Cross-sectional Study. ACR Open Rheumatol. 2020;2:478-490. doi: 10.1002/acr2.11163

[56] Dunphy MP, Freiman A, Larson SM, Strauss HW. Association of vascular 18F-FDG uptake with vascular calcification. J Nucl Med. 2005;46:1278-84.

[57] Ben-Haim S, Kupzov E, Tamir A, Israel O. Evaluation of 18F-FDG uptake and arterial wall calcifications using 18F-FDG PET/CT. J Nucl Med. 2004;45:1816-21.

Line 275: 3 references are not correct regarding text and numbers after extensive text revision.

We thank the reviewer for noticing this mistake that has now been corrected.
